# GROUNDING ALEATORIC UNCERTAINTY IN UNSUPERVISED ENVIRONMENT DESIGN

## ABSTRACT

In reinforcement learning (RL), adaptive curricula have proven highly effective for learning policies that generalize well under a wide variety of changes to the environment. Recently, the framework of Unsupervised Environment Design (UED) generalized notions of curricula for RL in terms of generating entire environments, leading to the development of new methods with robust minimax-regret properties. However, in partially-observable or stochastic settings (those featuring *aleatoric uncertainty*), optimal policies may depend on the ground-truth distribution over the aleatoric features of the environment. Such settings are potentially problematic for curriculum learning, which necessarily shifts the environment distribution used during training with respect to the fixed ground-truth distribution in the intended deployment environment. We formalize this phenomenon as *curriculum-induced covariate shift*, and describe how, when the distribution shift occurs over such aleatoric environment parameters, it can lead to learning suboptimal policies. We then propose a method which, given black-box access to a simulator, corrects this resultant bias by aligning the advantage estimates to the ground-truth distribution over aleatoric parameters. This approach leads to a minimax-regret UED method, SAMPLR, with Bayes-optimal guarantees.

## 1 INTRODUCTION

Adaptive curricula, which dynamically adjust the distribution of training environments to optimize the performance of the resulting policy, have played a key role in many recent achievements in deep reinforcement learning (RL). Applications have spanned both single-agent RL (Portelas et al., 2020; Wang et al., 2019; Zhong et al., 2020; Justesen et al., 2018), where adaptation occurs over environment variations, and multi-agent RL (MARL), where adaptation can additionally occur over co-players (Silver et al., 2016; Vinyals et al., 2019; Stooke et al., 2021). By presenting the agent with challenges at the threshold of its abilities, such methods demonstrably improve the sample efficiency and the generality of the final policy (Matiisen et al., 2017; Dennis et al., 2020; Jiang et al., 2021b;a).

This work introduces a fundamental problem relevant to adaptive curriculum learning methods for RL, which we call *curriculum-induced covariate shift* (CICS). Analogous to the covariate shift that occurs in supervised learning (SL), CICS refers to a mismatch between the input distribution at training and test time, and in this case, specifically when the distribution shift is caused by the selective sampling performed by an adaptive curriculum. While there may be cases in which CICS impacts model performance in SL, adaptive curricula for SL have generally not been found to be as impactful as in RL (Wu et al., 2021). Therefore, here we focus on addressing this problem specifically as it arises in the RL setting, and leave investigation of its potential impact in SL to future work.

To establish precise language around adaptive curricula, we cast our discussion under the lens of Unsupervised Environment Design (UED, Dennis et al., 2020). UED provides a formal problem description for which curriculum learning is the solution, by defining the *Underspecified POMDP* (UPOMDP; see Section 2), which expands the classic POMDP with a set of *free parameters* $\Theta$, representing the dimensions along which the environment may vary across episodes. The goal of UED is then to adapt distributions over $\Theta$, so to maximize some objective, which could be tied to an RL agent's performance over this distribution. This allows us to view adaptive curricula as emerging via a multi-agent game between a *teacher* that proposes environments with parameters $\theta \sim \Theta$ and a *student* that learns to solve them. In addition to the notational clarity it provides, this formalism lends

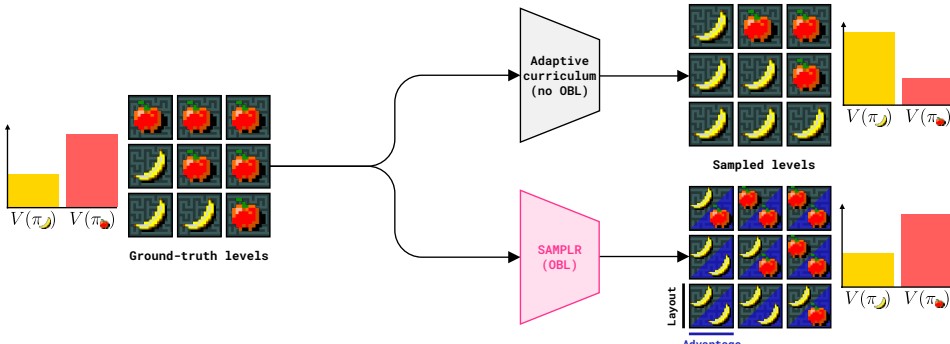

Figure 1: Adaptive curricula can result in covariate shifts in environment parameters with respect to a fixed ground-truth distribution $\overline{P}(\theta)$ (see top path), e.g. whether apple or banana is the correct fruit to choose per level. Here, the policy $\pi_{🍎}$ ($\pi_🍌$) always chooses apple (banana). Our method, SAMPLR (bottom path) matches the advantages in sampled levels to the advantages observed if sampling levels from $\overline{P}(\theta)$ (blue triangles), thus constraining the optimal policy under the curriculum distribution $P(\theta)$ to match that under $\overline{P}(\theta)$.

the analysis of adaptive curricula to useful game theoretic constructs, such as Nash equilibria (NE, Nash et al., 1950).

This game-theoretic view has led to the development of curriculum methods with principled robustness guarantees, such as PAIRED (Dennis et al., 2020) and Prioritized Level Replay (PLR, Jiang et al., 2021a), which showed that curricula optimizing for the student's regret lead to minimax regret (Savage, 1951) policies at Nash equilibria, implying the agent can solve all solvable environments within the training domain. While other methods can also be cast in this framework, they do not hold the same desirable property at equilibrium. For this reason, we will focus on addressing CICS for regret-maximizing UED, but note that our solution can be used with other UED methods.

To see how the CICS can be problematic, consider the simplified case of training a self-driving car in simulation, so it learns to take the fastest route from home to office. Suppose traffic data shows on 70% of the days, Route 1 is faster than Route 2. Moreover, on any given day the self-driving car cannot infer which route is faster ahead of time, so always picking Route 1 is faster in expectation. To support training a policy using an adaptive curriculum, one could build a simulator which sets road conditions per episode based on a random day sampled from the traffic data. However, adaptive curriculum over the traffic settings may oversample days when Route 1 is closed—perhaps because it finds the agent needs more practice on Route 2—shifting the best choice to Route 2 in training. In fact, methods for minimax regret UED like PLR would keep shifting the distribution of fastest route, to maximize the agent's regret. Their curriculum dynamics would map to the zero-sum game of matching pennies, in which one player wins for guessing whether the other chose heads or tails; the NE corresponds to each player randomly playing each option half the time. Randomly picking the route in this way is suboptimal, because *in reality* Route 1 is faster in expectation. This example is depicted in Figure 1, where the two routes are replaced by an apple and banana.

If, on a given day, the faster route could be identified before having to pick one, the agent could choose optimally. Instead, it is an aleatoric parameter, inducing irreducible uncertainty in the limit of infinite experiential data (Der Kiureghian & Ditlevsen, 2009). When CICS occurs over such parameters, with respect to a ground-truth distribution of environments $\overline{P}(\theta)$, the learned policy can be suboptimal with respect to $\overline{P}$. It can therefore be useful to *ground*—that is, to constrain—the aleatoric parameters $\Theta' \subset \Theta$ to $\overline{P}(\theta')$ when it is known or can be learned, as in simulation or from real-world data. However, grounding all $\theta$ is undesirable, preventing the curriculum from sampling enough opportunities to learn from useful scenarios with low support under $\overline{P}(\theta)$.

In this work, we formalize the problem of CICS in RL, and provide a solution by proposing a UED method to find robustly Bayes-optimal policies where $\theta'$ is grounded to $\overline{P}(\theta')$. Our solution called Sample-Matched PLR (SAMPLR) extends PLR, a state-of-the-art UED algorithm, by constraining the advantage estimates to match those observed if training under $\overline{P}(\theta')$. This advantage correction adapts Off-Belief Learning (Hu et al., 2021) from cooperative MARL, revealing an intriguing connection between curriculum biases observed in single and multi-agent RL. Our experiments in challenging

environments based on the NetHack Learning Environment (NLE, Küttler et al., 2020) demonstrate that SAMPLR learns near-optimal policies under CICS, in cases where standard PLR fails.

## 2 BACKGROUND

### 2.1 UNSUPERVISED ENVIRONMENT DESIGN

The problem of Unsupervised Environment Design (UED, Dennis et al. (2020)) is the problem of automatically generating an adaptive distribution of environments which will lead to policies that successfully transfer within a target domain. The domain of possible environments is represented by an Underspecified POMDP (UPOMDP), which adds a set of free parameters to the standard definition of a POMDP, along which each concrete instantiation, or *level*, of the UPOMDP. For instance, these free parameters can be the position of obstacles in a maze, or friction coefficients in a physics-based task. Formally a UPOMDP is defined as a tuple $\mathcal{M} = \langle A, O, \Theta, \mathcal{S}, \mathcal{T}, \mathcal{I}, \mathcal{R}, \gamma \rangle$, where $A$ is a set of actions, $O$ is a set of observations, $\Theta$ is a set of free parameters, $S$ is a set of states, $\mathcal{T} : S \times A \times \Theta \rightarrow \boldsymbol{\Delta}(S)$ is a transition function, $\mathcal{I} : S \rightarrow O$ is an observation (or inspection) function, $\mathcal{R} : S \rightarrow \mathbb{R}$ is a reward function, and $\gamma$ is a discount factor. UED typically approaches the curriculum design problem as training a *teacher* agent that co-evolves an adversarial curriculum for a *student* agent, for example, by maximizing the student's regret.

We will focus on a recent UED algorithm called Robust Prioritized Level Replay (PLR$^\perp$, Jiang et al., 2021b), which performs environment design via random search. PLR maintains a buffer of the most useful levels for training, according to some learning potential score—typically based on a regret approximation, such as the positive value loss—and with probability $p$, actively samples the next training level from this level buffer instead of the ground-truth training distribution. This selective-sampling mechanism has been demonstrated to greatly improve sample-efficiency and generalization in several domains, while provably leading to a minimax regret policy for the student at NE. In maximizing regret, PLR curricula naturally avoid unsolvable levels, which have no regret.

### 2.2 OFF-BELIEF LEARNING

In cooperative MARL, self-play promotes the formation of cryptic conventions—arbitrary sequences of actions that allow agents to communicate information about the environment state. These conventions are learned jointly among all agents during training, but are arbitrary and hence, indecipherable to independently-trained agents or humans at test time. Crucially, this leads to policies that fail to perform zero-shot coordination (ZSC, Hu et al., 2020), where independently-trained agents must cooperate successfully without additional learning steps, or ad-hoc team play. Off-Belief Learning (OBL) resolves this problem by forcing agents to assume their co-players act according to a fixed, known policy $\pi_0$ until the current time $t$, and optimally afterwards, conditioned on this assumption. If $\pi_0$ is playing uniformly random, this removes the possibility of forming arbitrary conventions.

Formally, let $G$ be a decentralized, partially-observable MDP (Dec-POMDP, Bernstein et al., 2002), with state $s$, joint action $a$, observation function $\mathcal{I}^i(s)$ for each player $i$, and transition function $\mathcal{T}(s, a)$. Let the historical trajectory $\tau = (s_1, a_1, ...a_{t-1}, s_t)$, and the action-observation history (AOH) for agent $i$ be $\tau^i = (\mathcal{I}^i(s_1), a_1, ..., a_{t-1}, \mathcal{I}^i(s_t))$. Further, let $\pi_0$ be an arbitrary policy, such as a uniformly random policy, and $\mathcal{B}_{\pi_0}(\tau | \tau^i) = P(\tau_t | \tau_t^i, \pi_0)$, a belief model predicting the current state, conditioned on the AOH of agent $i$ and the assumption of co-players playing policy $\pi_0$ until the current time $t$, and optimally according to $\pi_1$ from $t$ and beyond. OBL aims to find the policy $\pi_1$ with the optimal, *counter-factual value function*,

$$V^{\pi_0 \rightarrow \pi_1}(\tau^i) = \mathbb{E}_{\tau \sim \mathcal{B}_{\pi_0}(\tau^i)} \left[ V^{\pi_1}(\tau) \right]. \tag{1}$$

As the agent conditions its policy on the realized AOH $\tau^i$, while transition dynamics are based on states sampled from $\mathcal{B}$, this mechanism is called a *fictitious transition*. In Section 5, we show how OBL's fictitious transition can be adapted to the single-agent curriculum learning setting to address CICS, by interpreting the curriculum designer in UED as a co-player.

## 3 RELATED WORK

The mismatch between training and testing distributions of input features is referred to as *covariate shift*, and has long served as a fundamental problem for the machine learning community. Covariate shifts have been extensively studied in supervised learning (Vapnik & Chervonenkis, 1971; Huang et al., 2006; Bickel et al., 2009; Arjovsky et al., 2019). In RL, prior works have largely focused on covariate shifts due to training on off-policy data (Sutton et al., 2016; Rowland et al., 2020; Espeholt et al., 2018; Hallak & Mannor, 2017; Gelada & Bellemare, 2019; Thomas & Brunskill, 2016) including the important case of learning from demonstrations (Pomerleau, 1988; Ross & Bagnell, 2010). Recent work also aims to learn invariant representations robust to covariate shifts (Zhang et al., 2019; 2021). More generally, CICS can be interpreted as a kind of sample-selection bias (Heckman, 1979). We believe this work to be the first to formalize and provide a solution to the problem of covariate shifts in reinforcement learning due to curriculum learning.

Our method fixes a critical flaw that can cause curricula to fail under CICS—an important problem as curricula have been shown to be essential for training RL agents across many of the most challenging domains, including combinatorial gridworlds (Zhong et al., 2020), Go (Silver et al., 2016), StarCraft 2 (Vinyals et al., 2019), and achieving comprehensive task mastery in open-ended environments (Stooke et al., 2021). While this work focuses on PLR, other recent methods include minimax adversarial curricula (Wang et al., 2019; 2020) and curricula based on changes in return (Matiisen et al., 2017; Portelas et al., 2020). Most similar to our work, OFFER (Ciosek & Whiteson, 2017) adapts a curriculum over transition functions and uses importance sampling to correct for biased gradient estimates. Unlike this work, Ciosek & Whiteson (2017) requires whitebox access to the transition function and does not directly study the impact of CICS on the learning dynamics. Curriculum methods have also been studied in goal-conditioned RL (Florensa et al., 2018; Campero et al., 2021; Sukhbaatar et al., 2018; OpenAI et al., 2021), though CICS does not occur here as goals are observed by the agent. Lastly, domain randomization (DR, Sadeghi & Levine, 2017; Peng et al., 2017) can be seen as a degenerate form of UED, though curriculum-based extensions of DR have also been studied (Jakobi, 1997; Tobin et al., 2017).

Prior work has also investigated methods for learning Bayes optimal policies under uncertainty about the task (Zintgraf et al., 2020; Osband et al., 2013), based on the framework of Bayes-adaptive MDPs (BAMDPs) (Bellman, 1956; Duff, 2002). In this setting, the agent can adapt to an unknown MDP over several episodes by acting to reduce its uncertainty about the identity of the MDP. In contrast, SAMPLR learns a robustly Bayes optimal policy for the case of zero-shot transfer. Further unlike these works, our setting assumes the distribution of certain aleatoric parameters are biased during training, which would lead to biased *a posteriori* uncertainty estimates with respect to the ground-truth distribution when optimizing for the BAMDP objective. Instead, SAMPLR proposes a means to correct for this bias assuming knowledge of the true environment parameters for each level, to which we can safely assume access in curriculum learning.

## 4 CURRICULUM-INDUCED COVARIATE SHIFT

As UED algorithms formulate curriculum learning as a multi-agent game between teacher and student agents, we can formalize when CICS become problematic by considering the equilibrium point of this game: Let $\theta$ be the environment parameters controlled by UED, $\overline{P}(\theta)$, the ground truth distribution of $\theta$, and $P(\theta)$, the curriculum distribution at equilibrium. Further, let $\tau_t$ be $(o_1, a_1, ..., a_{t-1}, o_t)$, the student agent's action-observation history (AOH) until time $t$ (though we will use simply $\tau$ when clear from context). The optimal action-value function $Q^*$ with respect to $\overline{P}(\theta)$ can then be expressed as a marginalization over $\theta$:

$$\overline{Q}^*(a_t|\tau_t) = \mathbb{E}_{\substack{\tau_{t+1:\infty} \\ a_{t+1:\infty} \\ \sim \pi^*}} \left[ \sum_{l=0}^{\infty} \gamma^l r_{t+l} \right] = \sum_{\theta} \overline{P}(\theta|\tau_t)\overline{Q}^*_{\overline{P}}(a_t|\tau_t, \theta) \tag{2}$$

$$\propto \sum_{\theta} \overline{P}(\theta)\overline{P}(\tau_t|\theta)\overline{Q}^*_{\overline{P}}(a_t|\tau_t, \theta). \tag{3}$$

From Equation 2, we see that $\overline{Q}^*(a_t|\tau_t)$ remains optimal under different values of $\overline{P}(\theta)$ at each time $t$ as long as it is possible to infer $\theta$ deterministically from $\tau_t$, implying that $\overline{P}(\theta'|\tau_t) = 1$ for some $\theta$,

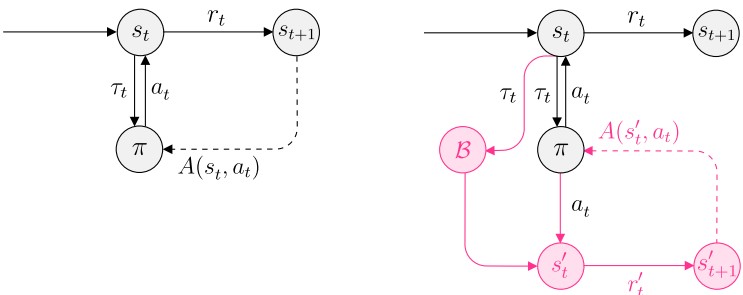

Figure 2: A standard RL transition (left) and the fictitious transition used by SAMPLR (right).

in which case the RHS of Equation 1 reduces to the LHS. If $\overline{P}(\theta|\tau_t) < 1$ for all $\theta$, then some subset $\theta' \subset \Theta$ results in irreducible uncertainty at time $t$, thereby constituting aleatoric parameters. Letting $Q^*(a_t|\tau_t)$ be the optimal action-value function when $\overline{P}(\theta)$ is replaced with $P(\theta)$ in Equation 2, we can then state that curriculum-induced covariate shift results in suboptimal policies with respect to the ground truth distribution when

$$\arg\max_a Q^*(a|\tau_t) \neq \arg\max_a \overline{Q}^*(a|\tau_t).$$

Moreover, this formulation highlights how this effect results from the presence of aleatoric parameters.

This description categorizes errors made by a policy trained on $P$ evaluated on levels drawn from $\overline{P}(\theta)$ into two categories: The first type of error, which we call a *mistake*, simply arises when $P(\theta'') = 0$ for some non-aleatoric $\theta''$ where $\overline{P}(\theta'') > 0$, i.e. the policy did not train on trajectories needed to learn to behave optimally under some set of otherwise identifiable level parameters. This is distinct from the second kind of error, which we call a *misunderstanding*, and which corresponds to the problematic mismatch between training and test time distributions of specifically aleatoric parameters $\theta'$, so that $P(\theta') \neq \overline{P}(\theta')$, as we just previously characterized.

This taxonomy also clarifies errors in cooperative MARL, where the co-players shape environment interactions, and thus play a similar role to the UED teacher. Failures in ZSC can then be diagnosed as due to (i) a mistake, because the co-players fail to generate trajectories that occur with test-time co-players; or (ii), a misunderstanding, because the train-time co-players shift the training distribution of aleatoric parameters $P(\theta')$, which impacts the inference of $\overline{P}(\tau) = \sum_{\theta'} \overline{P}(\theta')\overline{P}(\tau|\theta')$ needed for optimal cooperation—for example, through the use of cryptic conventions. This view then connects generalization errors in cooperative MARL to those in single-agent RL, implying that methods like OBL devised to solve one type of error in one of the settings may be adapted for the other. Indeed, we now describe our method, which does exactly this: By adapting OBL to single-agent curriculum learning, we can ground the values of $\overline{P}(\theta'|\tau_t)$ by forcing $P(\theta') = \overline{P}(\theta')$, thereby ensuring that the distribution of the aleatoric parameters at equilibrium is equivalent to their ground truth distribution.

## 5 SAMPLE-MATCHED PLR (SAMPLR)

We now describe how OBL's fictitious transition can be adapted for PLR$^\perp$ (Jiang et al., 2021a) to address CICS, resulting in Sample-Matched PLR (SAMPLR). To avoid CICS, we must ground $\theta'$ to the ground-truth distribution $\overline{P}(\theta')$, while allowing the remaining parameters in $\Theta$ to vary under UED, so as to still benefit from a curriculum. To achieve this, we adapt the fictitious transition to single-agent curriculum learning by treating the UED teacher as a co-player—one that performs the single action of choosing the level $\theta$ at the start of each episode, and subsequently performs no-ops. Under this fictitious transition, we ground the teacher's choice of $\theta$ such that the aleatoric parameters $\theta'$ are assumed to be sampled from $\overline{P}(\theta)$.

Thus, at each time $t$, the agent takes actions $a_t$ based on its AOH as usual, but estimates the advantage $A(a_t, s_t)$ using fictitious transitions, which assume subsequent state transitions and rewards occur with $\Theta'$ fixed to $\theta' \sim \overline{P}(\theta')$, sampled at the start of the episode. More formally, the fictitious transition is performed as $s'_t \sim \mathcal{B}(s'_t|\tau)$, $a_t \sim \pi(\cdot|\tau)$, $s'_{t+1} = \mathcal{T}(s'_t, a_t)$, and $r'_t = \mathcal{R}(s_{t+1})$, where $\tau$ is the

---

**Algorithm 1:** Sample-Matched PLR (SAMPLR)

---

Randomly initialize policy $\pi(\phi)$, an empty level buffer $\mathbf{\Lambda}$ of size $K$, and belief model $\mathcal{B}(s_t|\tau)$.

**while** *not converged* **do**

    Sample replay-decision Bernoulli, $d \sim \overline{P}_D(d)$

    **if** $d = 0$ or $|\Lambda| = 0$ **then**

        Sample level $\theta$ from level generator

        Collect $\pi$'s trajectory $\tau$ on $\theta$, with a stop-gradient $\phi_\perp$

        Use observed ground-truth states to update $\mathcal{B}$

    **else**

        Use PLR to sample a replay level from the level store, $\theta \sim \mathbf{\Lambda}$

        Collect fictitious trajectory $\tau'$ on $\theta$, based on $s'_t \sim \mathcal{B}$

        Update $\pi$ with rewards $\boldsymbol{R}(\tau')$

    **end**

    Compute PLR score, $S = \mathbf{score}(\tau', \pi)$

    Update $\mathbf{\Lambda}$ with $\theta$ using score $S$

**end**

---

AOH of the student. Figure 2 summarizes this transition mechanism. Here, the belief model $\mathcal{B}(s_t|\tau)$ can be expressed as

$$\mathcal{B}(s_t|\tau) = \sum_{\theta'} \overline{P}(s_t|\tau, \theta')\overline{P}(\theta'|\tau). \tag{4}$$

This shows that, assuming blackbox simulator access, we can generally implement $\mathcal{B}$ as follows: Periodically during training, we sample $N$ trajectories, such that each is generated under a level with $\theta' \sim \overline{P}(\theta')$. We use these $\{(\theta'_k, \tau_k)\}_{k=1}^{N}$ pairs to update a posterior model $\overline{P}(\theta'|\tau)$ that predicts the underlying $\theta'$ given $\tau$. We can then sample from $\mathcal{B}$ by first sampling $\theta \sim \overline{P}(\theta'|\tau)$, followed by stepping forward a parallel simulator that has been initially reset to the current AOH $\tau$, with fixed $\Theta' = \theta'$, thereby yielding a desired sample of $s_t$ according to $\mathcal{B}$.

In practice, it is often the case that $\theta'$ can be uniquely identified by some revelatory event by time $t$, so that $\overline{P}(\theta'|\tau) = 1$ for some $\theta'$ and $\tau$, and $\overline{P}(s_t|\tau, \theta') = \overline{P}(s_t|\tau)$ otherwise. In this case, we can implement the fictitious transition by setting $\theta' \sim \overline{P}(\theta')$ at the start of each episode; subsequent transitions will then be consistent with $\mathcal{B}$. For example, in the fruit choice example, whether apple or banana was the right goal is deterministically revealed by the final reward, and otherwise, does not impact transition dynamics. Additionally, we often only have limited access to $\overline{P}(\theta)$ throughout training, for example, if sampling $\overline{P}(\theta)$ is costly. In this case, we can learn an estimate $\hat{P}(\theta')$ using the samples we do collect from $\overline{P}(\theta)$, which can occur online. We then use $\hat{P}(\theta')$ in sampling fictitious transitions during UED. We refer to the resulting $\hat{P}(\theta')$ as a *learned belief prior*.

SAMPLR, summarized in Algorithm 1, incorporates this fictitious transition by replacing the advantages of trajectories on replay levels sampled by PLR$^\perp$ with their fictitious counterparts, as only these trajectories are used by PLR$^\perp$ for training. To reduce the cost of sampling $\overline{P}(\theta')$, we can use the new levels regularly sampled by PLR$^\perp$ to estimate the learned belief prior $\overline{P}(\theta')$, and use $\overline{P}(\theta')$ in sampling fictitious transitions on replay levels.

## 6   Grounded Policies Are Robustly Bayes Optimal

We can view OBL-based correction as a method for training a policy to be optimal with respect to the ground-truth value function, with levels sampled from some generating distribution $\Lambda$ defined as:

$$\overline{V}^{\Lambda}(\pi) = \mathbb{E}_{\tau \sim \mathcal{M}^{\Lambda}(\pi)}\left[\overline{V}^{\pi}(\tau)\right]. \tag{5}$$

Note that when $\Lambda = \overline{P}(\theta)$ this reduces to the ground-truth value function notated simply as $\overline{V}(\pi)$. First, we will note that, for any UED method, our OBL-based correction will ensure that, in equilibrium, the resulting policy is Bayes-optimal on the ground truth beliefs, on any trajectory sampled from $\mathcal{M}^{\Lambda}(\pi)$, the distribution of trajectories of $\pi$ in levels sampled from $\Lambda$.

**Remark 1.** *If $\pi$ is optimal with respect to the grounded value function $\overline{V}^{\Lambda}(\pi)$ then it is Bayes optimal with respect to the ground-truth parameter distribution $\overline{P}(\theta)$ on the support of $\mathcal{M}^{\Lambda}(\pi)$.*

*Proof.* By definition we have that $\pi \in \arg\max_{\pi \in \Pi}\{\overline{V}^{\Lambda}(\pi)\} = \arg\max_{\pi \in \Pi}\{\mathbb{E}_{\tau \sim \mathcal{M}^{\Lambda}(\pi)}\left[\overline{V}^{\pi}(\tau)\right]\}$. Since $\pi$ can condition on the initial trajectory $\tau$, the action selected after each trajectory can be independently optimized. Thus we have, for all $\tau \in \mathcal{M}^{\Lambda}(\pi)$, $\pi \in \arg\max_{\pi \in \Pi}\{\overline{V}^{\pi}(\tau)\}$ implying that $\pi$ is the optimal grounded policy and $\overline{V}^{\pi} = \overline{V}^{*}$. $\qquad\square$

Thus, assuming the base RL algorithm finds Bayes optimal policies, a UED method that optimizes the grounded value function, as done by SAMPLR, will result in Bayes optimal performance over the ground-truth distribution. When the UED method aims to maximize worst-case regret, we can prove an even stronger property we call *robust $\epsilon$-Bayes optimality*.

Let $\overline{V}^{\theta}(\pi)$ be the value function for $\pi$ evaluated on a specific level $\theta$. We will say that a policy is robustly $\epsilon$-Bayes optimal iff for all $\theta$ in the domain of $\overline{P}(\theta)$ and for all $\pi'$ we have

$$\overline{V}^{\theta}(\pi) \geq \overline{V}^{\theta}(\pi') - \epsilon.$$

Note how this differs from being only $\epsilon$-Bayes optimal, which means for all $\pi'$,

$$\overline{V}(\pi) \geq \overline{V}(\pi') - \epsilon$$

With robust $\epsilon$-Bayes optimality, we must be $\epsilon$-optimal even on levels which are rarely sampled from the ground-truth distribution. We will show that if SAMPLR is in an $\epsilon$-Nash Equilibrium, then a policy is robustly $\epsilon$-Bayes optimal with respect to the grounded value function $\overline{V}(\pi)$ rather than only $\epsilon$-Bayes optimal as one would expect from training directly on the true distribution of levels.

**Theorem 1.** *If $\pi$ is $\epsilon$-Bayes optimal by $\overline{V}^{\Lambda}(\pi)$ for $\Lambda$ minimizing worst-case regret as is done in SAMPLR, then it is robustly $\epsilon$-Bayes optimal with respect to the grounded value function, $\overline{V}(\pi)$.*

*Proof.* Let $\pi$ be $\epsilon$-optimal with respect to $\overline{V}^{\Lambda}(\pi)$ where $\Lambda$ minimizing worst-case regret with respect to $\pi$. Let $\overline{\pi}^{*}$ be an optimal grounded policy, and let $\theta$ be arbitrary. Then we have:

$$\overline{V}^{\theta}(\overline{\pi}^{*}) - \overline{V}^{\theta}(\pi) \leq \overline{V}^{\Lambda}(\overline{\pi}^{*}) - \overline{V}^{\Lambda}(\pi) \leq \epsilon \qquad (6)$$

Where the first inequality follows from $\Lambda$ minimizing worst-case regret with respect to $\pi$, and the second follows from $\pi$ being $\epsilon$-optimal on $\Lambda$. Rearranging terms gives the desired condition. $\qquad\square$

# 7 EXPERIMENTS

We investigate the performance of SAMPLR with respect to the the standard PLR and domain randomization in environments based on MiniHack (Samvelyan et al., 2021), a library for creating custom environments based on the runtime of the NetHack Learning Environment (NLE) (Küttler et al., 2020). Acting optimally in our environments requires grounding to the ground-truth distribution.

Our agents are trained using PPO (Schulman et al., 2017), using the best hyperparameters found via grid search, and use the policy architecture in Küttler et al. (2020). We tune the PLR-specific hyperparameters shared among PLR$^{\perp}$ and SAMPLR variants for each environment, based on the performance of PLR$^{\perp}$. Full details of our environments, agent architecture, and hyperparameters are provided in Appendix A, and our SAMPLR implementation, in Appendix B. We compare both standard SAMPLR and a variant called LP-SAMPLR that learns a belief prior over the true goal, against PLR$^{\perp}$ and standard PPO baselines. These baselines allows us to separate the relative changes in performance metrics due to curriculum learning and our proposed correction for CICS.

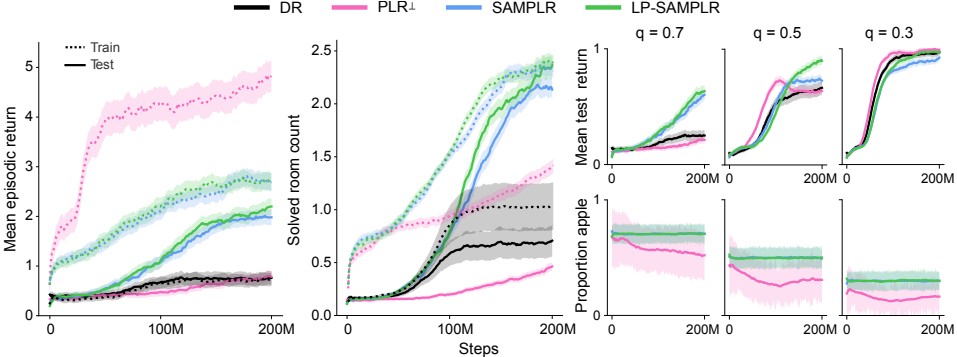

Figure 4: Episodic returns (left) and number of rooms in solved levels (middle) during training (dotted lines) and test on the ground-truth distribution (solid lines), for $q = 0.7$. Normalized test returns and proportion of apple goals during training for various $q$ are shown on the right. Plots show mean and standard error of 10 runs.

In our experiments, we first adapt the apple and banana example, depicted in Figure 3 into a fully procedurally-generated RL environment set in the world of NetHack. For each level, the agent can only learn to choose optimally in expectation at test time by grounding to the distribution of correct choices in the deployment domain. Specifically, in each level, the agent must traverse between one to eight randomly generated rooms, and in the final room, the agent must choose to eat the apple or the banana. The correct choice is fixed for each level, but indiscernible to the agent. Thus, the identity of the true goal acts as the aleatoric parameter. Figure 3 shows example levels from this environment.

This environment presents a hard exploration challenge for standard RL algorithms, as it requires learning to both navigate multiple rooms, as well as the NLE-specific skills of kicking doors and eating. The doors opening into adjacent rooms are locked. In order to go from one room to the next, the agent must learn to, potentially repeatedly, kick the locked door until it opens. Likewise, upon reaching a piece of fruit in the final room, the agent must learn to deliberately choose to eat the fruit. If the right choice of fruit were determinable per episode, we expect PLR$^\perp$'s adaptive curriculum to improve learning by selectively sampling levels at the threshold of the agent's abilities.

Let $\pi_A$ be the policy in which the agent always chooses the apple, and $\pi_B$, the banana. If the probability of the goal being apple $\overline{P}(A) = q$, the expected return of the agent is $R_A q$ under $\pi_A$ and $R_B(1 - q)$ under $\pi_B$. The optimal policy is then to act according to $\pi_A$ when $q > R_B/(R_A + R_B)$, and according to $\pi_B$ otherwise. We expect training under domain randomization, which samples each level completely at random from the ground-truth distribution $\overline{P}(\theta)$, defined by the environment parameterization, would converge to the correct choice of $\pi_A$ or $\pi_B$, assuming the environment is learnable by the choice of RL algorithm. In contrast, PLR evolves an adversarial curriculum, incentivizing PLR to shift the distribution over goals throughout training. For example, PLR is incentivized to flip the curriculum distribution to favor eating apples whenever the agent begins to consistently succeed at eating bananas. We thus expect the PLR curriculum to continually oscillate the preferred choice of goal.

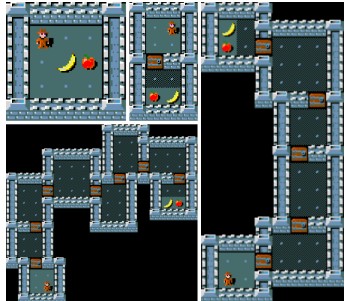

Figure 3: Levels from the stochastic fruit-choice environment. Across levels, the correct choice of fruit is distributed according to $\overline{P}$.

In our experiments, we set $R_A = 3$, $R_B = 10$, and $q$ in $\{0.7, 0.5, 0.3\}$, making it always optimal to follow the banana-eating policy $\pi_B$, but with the marginal benefit of doing so varying with $q$. We report train and test performance of each agent over 200M training steps in Figure 4. We find that DR struggles to learn an effective policy, plateauing at an expected return under $1.0$; PLR performs even worse, due to its adversarial curriculum shifting the distribution over the correct goal, leading to rapid oscillations in the optimal choice of fruit under the curriculum distribution, as visible in the high-variance oscillations in the proportion of apple goals selected by PLR for each value of $q$ in 4). This makes it difficult for the agent to settle on the optimal policy with respect to any ground-truth distribution. The proportion of each choice outcome, shown in Figure 5, reveals that both DR and PLR policies fail to eat any fruit most of the time, even after 200M training steps. In contrast, both SAMPLR and LP-SAMPLR realize a marked improvement in test performance by grounding PLR's

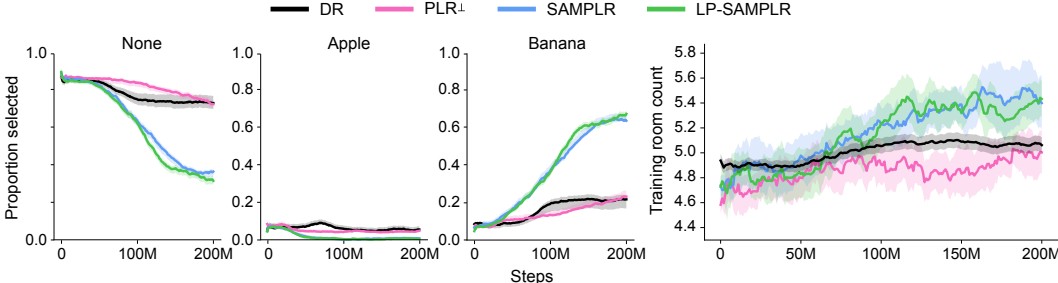

Figure 5: Left: Proportion of training episodes in which the agent fails to eat any fruit; eats the apple; or eats the banana. Right: Number of rooms in levels during training. Plots show mean and standard error of 10 runs.

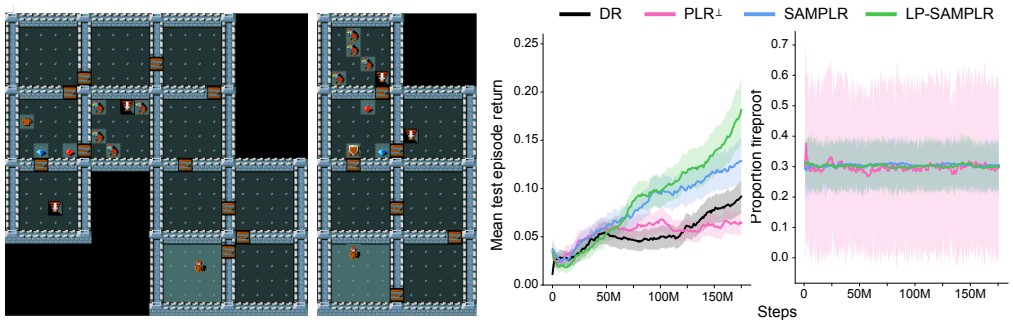

Figure 6: Left: Levels from FireDungeon. Middle: Test return on the ground-truth distribution of FireDungeon. Right: Proportion of training levels with fireproof armor. Plots show mean and standard error of 10 runs.

otherwise wild shifts in $q$. Figure 4 shows this improvement is most pronounced when the expected difference in $\pi_A$ and $\pi_B$ is smaller, and therefore easier for PLR to flip the agent's policy. Moreover, we see in Figure 5 that both SAMPLR methods present levels with higher room counts on average, indicating that early in training, it is able to discover easier, few-room levels in which the agent can capture reward signal, and which can then be made more complex to push the agent along its threshold of abilities.

We next turn to a more challenging environment that introduces additional NetHack-specific dynamics. The FireDungeon environment (see Figure 6 for example levels), requires the agent to navigate through up to 13 chambers, and ultimately choose between chamber A or B, each containing a valid goal. Reaching this goal ends the episode and provides the agent with a sparse reward. Further, in this penultimate chamber, there is an armor, which if worn grants the agent with fire resistance with probability $q$. Chamber B is marked by a red gemstone by the door and contains enemy units, whose fire attack will instantly kill the agent. Killing each enemy grants the agent with $+1$ final reward, which is only provided upon reaching either goal. The agent thus stands to attain a higher reward by attacking the enemies in Chamber B, before reaching the goal, only if the armor is fireproof. This environment presents an even more difficult exploration problem for the agent, yet we see in Figure 6, both SAMPLR variants begin to learn to solve this environment with significantly greater sample-efficiency than DR, while PLR again struggles to learn. As in the case of the previous stochastic choice environment, we see that PLR rapidly oscillates the key aleatoric parameter, which in this case, determines whether the armor is fireproof.

## 8 CONCLUSION

Using the formal notions of environment parameterizations in the framework of UED, we defined the problem of curriculum-induced covariate shift in RL. Our definition highlights the issues that can arise when there is persistent uncertainty over the environment parameters, either because the uncertainty is irreducible or because reducing the uncertainty is costly. We then adapted a fictitious transition mechanism previously used to improve zero-shot coordination in cooperative MARL to correct for this covariate shift. We demonstrated that our resulting algorithm, SAMPLR, avoids the pitfalls of this type of covariate shift, while preserving the benefits of curriculum learning.

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
