# OpenReview forum: "Grounding Aleatoric Uncertainty in Unsupervised Environment Design"
_ICLR.cc/2022/Conference — ICLR 2022 Submitted_

### Official Review · Reviewer_ZRvM · 2021-11-02

**Correctness:** 4
**Technical Novelty And Significance:** 2
**Empirical Novelty And Significance:** 2
**Recommendation:** 5
**Confidence:** 4

**Main Review:**

1. I am not sure what the "unsupervised environment design" in the title has got to do with the paper. The proposed method only selects/corrects for a distribution over a known set of environments.

2. I am really confused about what is the formal problem statement that the method is trying to solve, and what are the assumptions and properties available to the agent. Having a clear formal statement with assumptions after the introduction could be really helpful.

3. Since the method builds so heavily upon the previous work on PLR, it is worth including a summary of it in the background.

4. What do super-scripts of $\mathcal M$ in the POMDP tuple stand for? They are not used anywhere else again.

5. Why is the observation function assumed to deterministically map states to observations? Why is reward only a function of state, and not (s,a,s')? How do these choices impact the method?

6. In the definition of $\mathcal B$, it seems like $\tau_t$ is used as both the trajectory till step $t$ and the state in $\tau$ at step $t$. I am not sure how to interpret this.

7. In eqn 1, what does it mean to have $\pi_0 \rightarrow \pi_1$ in superscript? I don't see this notation being used like this anywhere else in the paper.

8. The paper introduces again introduces a new terminology and additional discussions on _mistake_ vs _misunderstanding_, but does not really use it again after. Further, I don't really see the difference, _mistake_ seems to be a sub-problem resulting due to _misunderstanding_, resulting from distribution mismatch where supports are not overlapping.

9. Paper uses "grounding X" everywhere, where X is some parameter. I am not sure what the authors mean by that.

10. One of the biggest drawbacks of the work is the assumption about access to the simulator. To perform the off-belief learning, the simulator is used in parallel to sample the next state transition under the true distribution of POMDP parameters instead of the distribution induced by the curriculum learning procedure.
 - a) The entire point of model-free methods is to not rely upon access to the simulator and just work with the data.
 - b) If the simulator/model is assumed to be known and accurate, then the comparisons should be made with other methods that leverage the simulator access as much as possible.
- c) Even when the simulator is known, how tractable is it to reset to a state corresponding to a particular trajectory's history under a different parameters of the POMDP, specifically when transition dynamics are stochastic? That is, how do you bring the parallel stochastic simulation to a state that corresponds exactly to observing that specific history?

11. It is later mentioned that "in practice, it is often the case the environment dynamics are uniquely determined by $\theta'$", and it is discussed how this can be exploited. However,  in this case, isn't the main motivation for aleatoric uncertainty as developed in eqn (2) and (3) non-existent?

12. Entire section 6 on Bayes optimality seems very uninformative. The arguments presented are only good for the tabular setting, which is probably not where one would be using curriculum learning or seeing the aleatoric uncertainly problem in practice. With function approximations you will need to additionally constrain policy comparison to policies with the function class, talk about optimality gap between the best policy in your class and $\pi^*$, and also discuss why it is reasonable to think that the proposed method will converge to $\epsilon$-Nash Equilibrium when using these function approximations.

13. While in the ideal setting the proposed method can address the covariate shift in the return _given_ a state, I am not sure how the distirbution shift in observing the specific states itself impacts the test performance. This is not a problem in tabular because states are not aliased, but in the function approximation setting the states that are visited often will tend to have a better policy as compared to the states which are visited less frequently. But if these less frequently visited states tend to get visited more frequently during test time, then there might be problems.


**Summary Of The Paper:**

Paper proposes a method to fix the covariate shift issue induced by curriculum learning RL methods when dealing with parameterized POMDPs. Specifically, curriculum learning methods choose a training distribution over the parameters of the POMDPs that might differ from the distribution that the agent might face during test-time, therefore the optimal solution under the distribution induced by the curriculum method might not be optimal under the test-time distribution. To fix this problem, the paper assumes access to the simulator such that given any previous trajectory the next state can be sampled under the true/test distribution, which is then used to update the policy.

**Summary Of The Review:**

I think the paper falls short because of strong assumptions made by the proposed method on simulators, uninformative theory, and poor presentation of the problem being addressed.

---

> ### Author Response · Authors · 2021-11-16
> **Responses to your detailed questions and helpful feedback (Part 2)**
>
> 9. In the second to last paragraph of page 2, we define our usage of "ground" to mean "to constrain the aleatoric parameters $\Theta' \subset \Theta$ to the ground truth distribution of $\theta'$. We have rephrased this passage to more clearly equate our usage of "to ground" with "to constrain" in this way.
>
> 10. We disagree that the proper point of comparison is to other methods that leverage the simulator as much as possible. The aim of this work is to address a fundamental flaw with curriculum learning methods such as PLR, making those the correct baselines. Further, we do not assume complete whitebox access to the simulator (for example, knowledge of the underlying transition matrix and states), but rather, only the ability to configure specific parameters controlling environment properties. We fully agree it is generally a good strategy to use the simulator as much as possible, and point out that methods like PLR already do this when we only have the ability to perform seeded resets of the simulator. Our method then fixes PLR under the curriculum-induced covariate shift when given the additional ability to set specific environment parameters. Are there other methods you have in mind that take more advantage of the simulator under the same assumptions on the simulator interface?
>
> Regarding (c), many (if not most) RL environments control the randomness with a specific environment pseudorandom-number generator seed. Therefore, resetting to the matching seed reduces the stochastic simulator to a deterministic one, making it trivial to reset the parallel simulator to the equivalent state.
>
> 11. You point out a subtlety here that we have clarified in the updated manuscript. Here "uniquely determined" refers to some aleatoric parameters $\theta’$ being uniquely identifiable at some action-observation history at time t collected on the environment instance. However, until $t$, the agent faces the same irreducible uncertainty of the value of $\theta'$, making it possible for the agent to make suboptimal decisions before $t$ that prevent the possibility of acting optimally—for example, deciding to enter the room with fire-spitting enemies when the armor the agent picked up was in fact not fireproof.
>
> 12. We agree that our theoretical discussion does not aim to prove convergence to an approximately Bayes optimal policy. In fact, in general, multi-agent RL, especially deep multi-agent RL, will have no such guarantees; this is well-documented in the community [1]. We have reworded Section 6 to make it clear to the reader that these limitations exist. We do not claim that SAMPLR by itself guarantees convergence to Bayes optimal policies, which is not the aim of our method—rather, if we were to use an RL algorithm that yields approximately Bayes optimal policies, SAMPLR will further guarantee that such policies will also be approximately Bayes optimal with respect to the ground-truth distribution of interest. We further show that for PLR, these Bayes optimal policies are “robustly Bayes optimal” in that they will yield an expected return on each level that is within $\epsilon$ of the expected return over the ground truth distribution of $\theta’$.
>
> 13. We agree that this is a possibility, though such suboptimality at test-time would be due to insufficient training on some specific set of states. This is a problem concerning insufficient frequency of training rather than the actual curriculum distribution, and should be addressed by ensuring previously visited environment instances will always have some chance of being resampled in the future, so that in the limit of infinite training, an adequate agent model should be able to learn the optimal behavior given an observation. Addressing special cases in which this problem occurs seems like a useful direction for future research. However, this problem is orthogonal to the one we address with our method, impacting both the OBL-corrected (SAMPLR) and non-OBL corrected (PLR) methods equally.
>
> Based on these additional clarifications, we hope you will consider increasing your score in support of this work. Otherwise, could you let us know what additionally needs to be done in your assessment to make this work ready for publication?
>
> References:
> [1] Eric Mazumdar, Lillian J Ratliff, and S Shankar Sastry. On gradient-based learning in continuous games. SIAM Journal on Mathematics of Data Science, 2(1):103–131, 2020

---

> ### Author Response · Authors · 2021-11-16
> **Responses to your detailed questions and helpful feedback (Part 1)**
>
> Thank you for your detailed feedback, which we believe will improve our final manuscript. Please see our responses to the helpful points raised in your review below:
>
> 1. As elaborated in Section 1 (Introduction), we cast our analysis of adaptive curricula for RL in the formal problem setting of Unsupervised Environment Design (UED). UED defines a problem setting (i.e. adapting the parameters of an environment, or more specifically, an underspecified POMDP, to maximize some metric of interest) for which adaptive curricula are the solution. This allows us to talk about adaptive curricula in a principled and exact way under an established problem setting and formalism.
>
> 2. The main problem we are solving is that curriculum learning methods can learn suboptimal policies under curriculum-induced covariate shifts. We provide an intuitive example of this problem in Section 1 and a formal characterization of this problem in Section 4. However, your comment makes clear that we should further emphasize how UED-based adaptive curricula are used to induce minimax regret policies, i.e. those induced by robust PLR. We will update the paper accordingly.
>
> 3. While we do provide a high-level summary of robust PLR in the second paragraph of Section 2.1 (Unsupervised Environment Design), we agree that more low-level details of robust PLR would benefit understanding and make the paper more self-contained. We will add the pseudocode for robust PLR in the appendix so readers can easily refer to the complete details of how PLR works. We will further emphasize in Section 2.1 that when the prioritization score of PLR is a regret estimate, PLR becomes a form of minimax regret UED.
>
> 4. Thanks for pointing out that these superscripts are superfluous. We have removed them.
>
> 5. Our method does not assume the observation function is deterministic, and in general, it can be a stochastic function mapping states to observations.
>
> 6. Throughout the paper, including Equation 4, \tau exclusively refers to the action-observation history (AOH) at time $t$: $\(o_1, a_1, ..., a_{t-1}, o_t \)$. We now emphasize this usage in the updated manuscript.
>
> 7. Equation 1 defines the counterfactual value function where co-players are assumed to play according to some fixed $\pi_0$ until time t, and optimally at time $t$ and beyond, according to an optimal policy $\pi_1$. In other words, this is the value function for an MDP that transitions according to the fictitious transition mechanism at each time $t$. This counterfactual value function (and arrow notation) was first introduced in Hu et al, 2021 (OBL). We have rephrased this section to emphasize how the arrow notation refers to the counterfactual value function assuming co-players transition from playing $\pi_0$ to $\pi_1$ at time $t$.
>
> 8. a. The intention of defining these two types of errors is to point out how our characterization of the curriculum-induced covariate shift clarifies two distinct classes of test time errors, where the second type, "misunderstandings," corresponds to harmful cases of the curriculum-induced covariate shift in both cooperative MARL and single-agent curriculum learning. This shared class of errors (misunderstandings) across these two problem settings is what motivates directly adapting OBL from cooperative MARL to the single-agent curriculum learning setting.
>
>    b. Thanks for pointing out this gray-area in the current definition of mistakes and misunderstandings, namely that a mistake in aleatoric parameters is also a form of misunderstanding. We have improved our definition based on this feedback by defining mistakes to be due to *zero* support in train distribution over a non-aleatoric (i.e. epistemic) parameter $\theta$ that appears at test time. The improved definition of mistakes, thanks to your feedback, now cleanly separates the cases of mistakes and misunderstandings.

---

### Official Review · Reviewer_R1Z8 · 2021-11-02

**Correctness:** 2
**Technical Novelty And Significance:** 2
**Empirical Novelty And Significance:** 2
**Recommendation:** 5
**Confidence:** 3

**Main Review:**

Strengths:

- This paper studies unsupervised environment design which is an intriguing and important problem to facilitate reinforcement learning with minimum human intervention.

- The proposed method consistently outperforms TLR and DR baselines in two discrete space task domains.

Weaknesses:

- The problem formulation does not seem to be entirely convincing to me. In this paper, an inference model $\bar{P}(\theta | \tau)$ is trained with the policy to predict the environment parameter $\theta$ given $\tau$. Given this belief model, why do the authors still stick to a Q value function as a marginalization over θ in Eqn. 2 instead of using $\bar{P}(\theta | \tau)$ to conduct online system identification and learns a policy $\pi(a | s, \theta)$ conditioned on the inferred $\theta$? Wouldn't that lead to better performance?

- The definition of the belief model in Eqn. 4 does not seem to be correct. Based on the Bayes' theorem, the RHS is supposed to be $\sum_{\theta'} \bar{P}(s_t | \tau, \theta') \bar{P}(\theta' | \tau)$. The marginalization is missing here.

- Several definitions are missing, which makes the paper hard to read. For example, how is $\mathcal{M}^{\Lambda}(\pi)$ defined? In Sec.2, $\mathcal{M}$ is defined as the tuple of UPOMDP but in Sec. 6 it seems to become a probability distribution. The definition of $\bar{Q}^*_{\bar{P}}(a_t | \tau_t, \theta)$ is also missing.

- Could the authors provide more discussions on how the fictitious trajectories help mitigate the curriculum-induced covariate shift? I do not think the explanation is sufficiently convincing in the current draft.

- The experiment setups seem to be a bit too simple. As shown in Figure 3 and Figure 6, the different goals are always placed next to each other in the same room. And there is no penalization for reaching the wrong goal. Therefore, the policies for solving different $\theta$ will only have minor differences. Specifically, the policy only needs to distinguish the goals in the last 1-2 steps, but the actions before that will be exactly the same. As a result, the learned Q value functions will be insensitive to the changes of $\theta$ except for the final state. Without evaluating the proposed method in a more challenging setup (e.g. different goals placed in different rooms, and negative rewards for reaching the wrong goal), it would be hard to justify the effectiveness of the proposed method.

- Although the authors claim that the proposed method generates curricula to facilitate the training of the policy, there is no quantitative or qualitative results that demonstrate what kind of curricula are generated. It would be better to include an analysis of how $\theta$ evolves during training and how it facilitates the training.

**Summary Of The Paper:**

This paper aim at generating curricula for training an RL agent to solve tasks sampled from an unknown distribution. Based on Prioritized Level Replay and Off-Belief Learning, an algorithm is proposed to train the policy using trajectories sampled from a learned belief model. The proposed method is evaluated in two goal-reaching tasks of discrete state and action spaces.

**Summary Of The Review:**

Although the paper tackles an important problem, the proposed technical solutions do not seem to be entirely convincing and the experiments are insufficient. I believe this paper needs another iteration before being ready for publication.

---

> ### Author Response · Authors · 2021-11-16
> **Responses to your helpful feedback**
>
> We appreciate the reviewer's useful feedback on our work, and further thank the reviewer for 	finding our work to successfully address an intriguing and important problem in RL (at least on two discrete control domains). Below, we address the reviewer's concerns in turn:
>
> - This method of system identification does not work when $\theta'$ has irreducible uncertainty. In such cases, including the various examples in the paper, the agent cannot effectively identify the system (e.g. whether Route 1 or Route 2 is faster) before committing to a course of action that is optimal for only a particular realization of $\theta'$. Our method allows the agent to get around this, by ensuring the policy learned is optimal in expectation over the ground-truth distribution of $\theta'$.
> - Thank you for spotting this typo. We corrected this expression in the updated manuscript.
> - Thanks also for spotting this missing definition. We agree this is a confusing shorthand that was not well defined in the paper. You are correct that $\mathcal{M}$ is simply the tuple for the UPOMDP. Here we mean for $\mathcal{M}^{\Lambda}(\pi)$ to simply mean a function that samples trajectories by following $\pi$ under instantiations of $\mathcal{M}$ sampled from $\Lambda$, where $\Lambda$ is some generating distribution of levels parameters $\theta$. We now clearly define the notation $\mathcal{M}^{\Lambda}(\pi)$ when it is first introduced (last sentence of page 6).
> - This is useful feedback for us to consider regarding the clarity of our explanation. The central argument is that the curriculum-induced covariate shift (CICS) over $\theta'$ is completely analogous to cryptic conventions in MARL, which are eliminated by grounding the co-player policies before time $t$ to a fixed, uniformly random policy using OBL's fictitious transition. This works because assuming trajectories up to the present (time $t$) were generated by co-players using a uniformly random policy means there is no useful signal in the specific action sequences for the agent's policy to condition on—eliminating the formation of cryptic conventions, which are arbitrary action sequences that encode state information. We then cast single-agent curriculum learning as a two-player game, where the teacher's sole action sets $\theta'$ at the beginning of the episode, according to samples drawn from the ground-truth distribution of $\theta'$ or some approximation thereof. This shows CICS can be similarly interpreted as a kind of cryptic convention between the teacher and student, and therefore OBL's fictitious transition will likewise resolve the issue. Please let us know if this clarified matters for you or if there are any additional questions.
> - The intention of this experiment is to showcase how CICS can result in state-of-the-art curriculum learning methods producing suboptimal policies, and how the fictitious transition used by SAMPLR fixes this issue. The experiments demonstrate this effectively. Placing additional goals in other rooms besides the last one would simply reduce to chaining multiple episodes of the current fruit-choice environment into a single episode, but the central environment dynamics and therefore CICS-based problem remains the same.
> Our problem settings clearly illustrate the value of our approach, as intended.
> - We agree that showing quantitative metrics demonstrating the structure of the emergent curricula is useful, which is why we show the number of rooms leading to the fruit choice in levels presented by each curriculum throughout the course of training (Figure 5, right). This metric is the natural measure of difficulty for this environment, and thus this plot shows the emergent complexity over level structure induced by the curriculum over time. We can clearly see in this figure that level difficulty increases much more rapidly under SAMPLR's curricula compared to DR (which remains approximately constant at 5 rooms as expected) and PLR, whose ability to induce an effective curriculum is significantly hindered by CICS.
>
> In considering these responses to your feedback, we hope you will consider increasing your score for our paper.

---

### Official Review · Reviewer_AAp9 · 2021-11-03

**Correctness:** 3
**Technical Novelty And Significance:** 3
**Empirical Novelty And Significance:** 3
**Recommendation:** 5
**Confidence:** 3

**Main Review:**

**Strengths:**

This work delivers what is promised experimentally. It tests the algorithm on NetHack and MiniHack and demonstrates that the proposed algorithm reduces the gap between training and test time evaluation.

**Weaknesses:**

There are a several issues with the clarity. I've expanded those comments in the clarity section of the review.

**Detailed Comments:**

C1: In figure 1 text you say ground-truth distribution $P(\theta)$ where in the main text you use $\bar{P}(\theta)$ for the ground-truth. I assume that the ground-truth is notated as $\bar{P}$, can you verify that this is correct and fix the figure 1?

C2: The belief is $P(\tau_t \bar \tau^i_t, \theta)$. What is $\tau_t$ since $\tau$ is $(s1, a1, s2, a2, \dots)$? Is the subscript index the state only? I know that this is inherited by off belief work but would be good to clarify the notation.

C3: What is the difference between $\bar{Q}$ and $Q$?

C4: "If $\bar{P}(\theta \dots)<1$ .. then some $\theta'$ result in irreducible uncertainty". Let's examine an example. Say we have $\theta_1$, $\theta_2$ and $\theta_3$. Say we have a trajectory $\tau$ for which the transitions and emissions where the same for all three environments but in the future all three environments end up in different states. Therefore, $\bar{P}(\theta \bar \tau_t)<1$ (since we can't tell which one of the parameters is for sure). Your claim is that this is aleatoric uncertainty however, you'll be able to reduce such uncertainty if you progress in the environment and acquire more evidence.

C5: You say "we see that $\bar{Q}*$ remains optimal ... as long as $\bar{P}(\theta' \bar \tau_t) = 1$" - Can you explain why is this the case? I can't see why you rely on $\bar{P}(\theta' \bar \tau_t)=1$ for the optimality argument.

C6: Shouldn't "if d = 0 then" be "if d = 0 or $\Lambda$ is empty" in algorithm 1?

C7: You say $\lambda=\bar{P}(\theta)$. Is $\Lambda$ a distribution or a parameter? Also what is $M^\Lambda$? Is it the simulator where you sample parameters from the distribution $\Lambda$? Is it fair to say that in the expectation in eq 5) you sample a $\theta \sim \Lambda$ and then play $\pi$ in the environment parametrized by $\theta$ and gather the trajectory $\tau$?

C8: Related to C7 but at some point you say $\bar{V}^\theta$ which confuses me. Is this an expectation like $\bar{V}^\Lambda$ ?

C9: In the experiments, can you clarify how many seeds were run and what's the shaded area means?


C10: Related work: [https://www.aaai.org/ocs/index.php/AAAI/AAAI17/paper/viewPaper/14378](https://www.aaai.org/ocs/index.php/AAAI/AAAI17/paper/viewPaper/14378) - this work is using importance sampling to fix the bias of sampling rare environments

**Summary Of The Paper:**

This work proposes a solution to the problem of covariate shift appeared in adaptive curriculum learning where the distribution of parameters of the environment at test time is different from the one at training time. This is caused in algorithms like PAIRED or PLR because the adaptive curriculum learning algorithm biases towards regret maximising environments and as such trains the learner on a biased distribution instead of the ground truth one. The authors propose a solution to the problem under the assumption that the designer knows the ground truth distribution of the parameters of the environment. The solution relies on using fictitious transitions to compute the value function, using Off Belief learning — an algorithm for cooperative MARL.

**Summary Of The Review:**

I'm suggesting marginal below acceptance threshold because I find certain parts of the paper unclear. The experiments support the argument of the paper which is great however I'd like to understand and see the paper improved in a few aspects:
- Understand if the notion of aleatoric parameters is the correct one here. See C4
- Understand C6 and C7 better in order to follow the complete argument and the proofs.

---

> ### Author Response · Authors · 2021-11-16
> **In response to your encouraging and helpful feedback**
>
> Thank you for your overall supportive review of our work. Moreover, thank you for pointing out several points of inclarity in the existing manuscript. We now address each of the points in your review in turn, and have adjusted the manuscript with the corresponding clarifications:
>
> - This is a well-spotted typo. The overline above the $P$ is indeed missing here.
> - Throughout the paper, $\tau_t$ (and equivalently, simply $\tau$) refers to the agent's action-observation history until time $t$, $\(o_1,a_1,...,a_{t-1},o_t\)$. This is first defined in the description of OBL in Section 2.2. We make it clearer that $\tau$ and $\tau_t$ are used synonymously in the paper.
> - Here, $\overline{Q}^*$ refers to the optimal action-value function under grounded distribution $\overline{P}(\theta)$, while $Q^*(\theta)$ refers to the optimal action-value function under some other distribution $P(\theta)$. This is defined in Section 4 (first paragraph of page 5).
> - This is a great comment pointing out an important subtlety: We refer to $\theta'$ as an aleatoric parameter if its uncertainty is irreducible at any time step $t$. Such aleatoric parameters only cause a problem when the action-value function $Q^*$ under the curriculum's covariate-shifted distribution differs from that, $\overline{Q}^*$ under the ground-truth distribution. That is, we care about whether a parameter's uncertainty is reducible or irreducible before the decision has to be made. For instance, if I'm betting on a coin toss in the next time step, or a coin that was tossed but is hidden from me until I bet, the decisions are effectively the same so it seems useful to refer to both as decisions under aleatoric uncertainty. We show that OBL effectively mitigates cases of covariate shift over such problematic $\theta'$. We have now clarified this point in Section 4.
> - If $\overline{P}(\theta'|\tau_t) = 1$, then the RHS in Equation 1 reduces to simply $\overline{Q}^*(a_t|\tau_t, \theta)$, which by definition is equivalent to the LHS. Similarly, if we replace $\overline{P}(\theta'|\tau_t)$ with another distribution $P(\theta'|\tau_t)$, since we infer with probability 1 the identity of $\theta'$, the RHS again reduces to $Q(a_t | \tau_t, \theta)$. Therefore, in this case, the optimal behavior conditioned on $\tau_t$ does not change if $\overline{P}$ is replaced by another distribution $\{P}$.
> - Yes, this is technically correct, though in practice $\overline{P}_D(d)$ is also a function of $\Lambda$, so it encapsulates this additional conditional logic in the case $|\Lambda|$ is under some minimum size.
> - Under this definition, $\Lambda$ is being equated with the distribution $\overline{P}(\theta)$. Your understanding of $\mathcal{M}^{\Lambda}$ is basically correct. Specifically, we mean for $\mathcal{M}^{\Lambda}(\pi)$ to refer to a function that samples trajectories by following $\pi$ Under instantiations of $\mathcal{M}$ sampled from $\Lambda$, where $\Lambda$ is some generating distribution of levels parameters $\theta$. We now clearly define the notation $\mathcal{M}^{\Lambda}(\pi)$ when it is first introduced (last sentence of page 6).
> - You are right that this notation is unclear. We now define $\overline{V}^{\theta}$ as the value function $\overline{V}^{\pi}$ conditioned on the UPOMDP parameters being set specifically to $\theta$, i.e. a particular level.
> - We have included these important experimental details: Each curve represents the mean and standard error over 10 experiment seeds.
> - Thank you for the pointer to OFFER. We have included it in the related works section. The difference between this method and ours is that they parameterize the environment via the parameters of the learned transition function, which definitionally assumes read access to the true transition function for each environment instance. This is an assumption that is not required by our method.
>
> In light of these clarifications, would you consider increasing your score for our paper? If not, could you let us know any additional changes you would like to see in order for this work to be accepted?

---

### Official Review · Reviewer_UQqT · 2021-11-04

**Correctness:** 3
**Technical Novelty And Significance:** 2
**Empirical Novelty And Significance:** 2
**Recommendation:** 5
**Confidence:** 2

**Main Review:**

The problem here appears to be that an adversarial curriculum approach is allowed to modify a stochastic parameter that is not observable at run-time by the agent, but drastically changes the optimal policy. I had difficulty understanding the authors' approach, which at first glance appears very complicated but somehow reduces to something extremely simple in their experiments. This might be due to their choice of rather simple 2D grid-world type environments. I believe they may be correct in identifying a problem in this type of approach, but I am not sure that this is novel.

I have some concerns with this paper:
- Adversarial curriculum learning over a parameter that completely changes the optimal policy seems like an obviously bad thing. The (Dennis et al, 2020) paper you cite even mentions in the abstract that minimax often leads to unsolvable environments. Can you not just exclude these manually?

- Their approach is introduced in sec 2.2 by analogy to "cryptic conventions" in cooperative MARL, which is unfortunately not very intuitive unless you have a background in this subject. Reading this, I am not sure this connection is needed or helpful.

- "In practice, it is often the case that environment dynamics are uniquely determined by θ′ so that
P(θ′|τ) = 1" - EDIT for clarity: The math indicates you mean there other way around - that the curriculum parameter can be uniquely determined by the trajectory. This might be correct in your simple fruit environment, but seems like a questionable assumption in general? One could move a box in the environment that does not impact the policy, or if you have random elements in the agent actions, noise in agent transitions, or observations, you might get similar trajectories using very different curriculum parameters. This will also complicate the modelling you make when learning \bar{P}(theta|tau), in the general case it seems like you might need a more complex density estimator than e.g. a regular Guassian model.

- The experiments use very simple environments. As I understand it, there is only one curriculum parameter in the fruit world?

Minor:
typo: "persisitent"

**Summary Of The Paper:**

The authors consider the problem of covariate shift in recent adaptive curriculum learning methods. They propose to address this by changing how the adversarial approach interacts with aleatoric (stochastic) parameters and demonstrate it on two 2D grid environments.


**Summary Of The Review:**

While automated curriculum learning is an important topic, the purported problem they are attacking seems a bit too obvious. I am not sure how novel this is, and their solution appears needlessly confusing and demonstrated only on toy examples.

---

> ### Author Response · Authors · 2021-11-16
> **Thank you for your helpful feedback**
>
> We thank the reviewer for their useful feedback, which points out details that we clarify in our updated manuscript. Please see our responses addressing the specific concerns below:
>
> > Adversarial curriculum learning over a parameter that completely changes the optimal policy seems like an obviously bad thing.
>
> Quite the opposite: There are settings where doing a form of UED over these parameters is highly desirable! Imagine a car racing environment, where there is a fixed (low) probability that a given patch of road has “black ice,” which makes the road more slippery and hence driving more difficult. If the probability of black ice is too low, the agent may not get enough experience to know how to robustly respond to these conditions. UED over the location of black ice patches allows the agent to experience states in which it must recover from black ice. However, using most UED algorithms in this setting would cause CICS, as such a curriculum might overly focus on black ice patches, making the agent overly conservative. Our method corrects future returns by grounding them to the ground-truth distribution over black ice patches, thereby avoiding this problem, while still allowing the curriculum to expose the agent to states from which it must recover from driving over black ice.
>
> > The (Dennis et al, 2020) paper you cite even mentions in the abstract that minimax often leads to unsolvable environments. Can you not just exclude these manually?
>
> SAMPLR is not addressing unsolvable environments; this is already taken care of by the UED algorithms on which we build on top, i.e. PLR. Hence explicitly filtering these out would not be useful in our setting. In general, minimax regret UED methods generate a regret-maximizing curriculum. Unsolvable levels have a max regret of 0, so the curriculum is disincentivized from generating such levels.
>
> > "In practice, it is often the case that environment dynamics are uniquely determined by θ′ so that P(θ′|τ) = 1" - EDIT for clarity: The math indicates you mean there other way around - that the curriculum parameter can be uniquely determined by the trajectory. This might be correct in your simple fruit environment, but seems like a questionable assumption in general? One could move a box in the environment that does not impact the policy, or if you have random elements in the agent actions, noise in agent transitions, or observations, you might get similar trajectories using very different curriculum parameters. This will also complicate the modelling you make when learning \bar{P}(theta|tau), in the general case it seems like you might need a more complex density estimator than e.g. a regular Guassian model.
>
> Thanks for pointing out that this wording is inaccurate. We have rephrased "environment dynamics are uniquely determined by $\theta'$" to "$\theta'$ is uniquely identifiable during the episode, so that $P(\theta'|\tau) = 1$ for some $\theta'$ and $\tau$." This specific passage describes the case where $\theta’$ can be uniquely identified by some revelatory event by some time $t$, and therefore by $\tau$ at $t$.
>
> You are correct that generally this assumption does not hold, and we need to resort to learning a posterior distribution $P(\theta|\tau)$ online during training using supervised learning or otherwise, have access to such a distribution to sample from the belief model. Our experiments focus on settings where this assumption does hold, so we can directly isolate the effect of our method—by assuming a correct belief model—on repairing the performance of curriculum learning methods like PLR under curriculum-induced covariate shifts.
>
> >  I believe they may be correct in identifying a problem in this type of approach, but I am not sure that this is novel.
>
> Finally, regarding the reviewer's concern about whether this is a novel problem, we have broadly reviewed the previous literature on addressing covariate shifts in ML, and found no prior work directly analyzing when curricula induce consequential covariate shifts in RL. We reiterate that to the best of our knowledge (based on our literature review), we believe we present the first work attempting to formally characterize how curricula can induce deleterious covariate shifts in RL. Our solution is novel, building off of a previously unnoticed connection between cooperative MARL and curriculum learning for single-agent RL.
>
> Given these clarifications, would you consider raising your score for our paper?

---

> > ### Comment · Reviewer_UQqT · 2021-11-24
> > **Response to Rebuttal**
> >
> > I thank the authors for their detailed responses, and the updates to the paper notation. The idea behind the paper seems interesting, although I think that only using experiments where estimating $p(\theta|\tau)$ is trivial really takes away from the empirical evaluation. However, the other reviewers also appear to have concerns with clarity of the notation, and upon reading the paper again I still think it has some serious issues. I still have trouble understanding the main theoretical underpinnings of how you unbias the policy gradients. To give some examples:
> >
> > 1) Starting already with "From Equation 2, we see that $\bar{Q}^*(a_t|\tau_t)$ remains optimal under different values of $\bar{P}(\theta)$ at each time $t$ as long as it is possible to infer $\theta$ deterministically from $\tau_t$, implying that $P(\theta'|\tau_t) = 1$ for some $\theta$,". First, this seems to indicate that this is a hard requirement of the approach (contrary to what was claimed in your reply above). Second,  Eq.2 is just making it explicit that $\theta$ was marginalized out of the trajectory distribution, I don't see how your claim about it having to be deterministically inferrable follows form this.
> >
> > 2) Eq. 3 is another reformulation but this is never even referred to in the text.
> >
> > 3) Are the Q functions really intended to be probability distributions on the left and right side of Eq.2, it's seemingly written as a regular expected-sum-reward in the middle of the formula? Finally, the different types of Q-functions used here are never defined.
> >
> > After that there is an analogy to cryptic conventions in MARL with a belief-space reformulation, but at its core the proposed unbiasing mechanic never seems to be clearly defined in one place. I understand that there is a parallel simulator running for doing policy updates with "fictitious transitions" as shown in Fig.2, but it is unclear what formula is actually used for the updates in the end. It would be helpful to have the proposed update rule and its mathematical motivation clearly explained in the beginning. As this will likely require a major revision, I cannot raise my score.

---

### Author Response · Authors · 2021-11-24
**Discussion**

As we enter the final discussion period, would the reviewers please take a look at our responses and consider updating their scores if their feedback has been adequately addressed?

---

### Decision · Program_Chairs · 2022-01-20

**Decision:**

Reject

**Comment:**

The paper tackles the problem of covariate shift in adaptive curriculum learning.  Unfortunately, the paper lacks clarity and the experiments are insufficient.  The author response clarified the notation and corrected many typos, however, the paper remains conceptually unclear as pointed out by the reviewers.  Hence this work is not ready for publication.